# Targeting NMDA Receptor Complex in Management of Epilepsy

**DOI:** 10.3390/ph15101297

**Published:** 2022-10-21

**Authors:** Shravan Sivakumar, Mehdi Ghasemi, Steven C. Schachter

**Affiliations:** 1Department of Neurology, University of Massachusetts Chan Medical School, Worcester, MA 01655, USA; 2Department of Neurology, Beth Israel Deaconess Medical Center, Boston, MA 02215, USA; 3Harvard Medical School, Boston, MA 02114, USA; 4Consortia for Improving Medicine with Innovation & Technology (CIMIT), Boston, MA 02114, USA

**Keywords:** *N*-methyl-D-aspartate (NMDA) receptor, seizure, epilepsy, NMDA receptor antagonist, clinical trial

## Abstract

*N*-methyl-D-aspartate receptors (NMDARs) are widely distributed in the central nervous system (CNS) and play critical roles in neuronal excitability in the CNS. Both clinical and preclinical studies have revealed that the abnormal expression or function of these receptors can underlie the pathophysiology of seizure disorders and epilepsy. Accordingly, NMDAR modulators have been shown to exert anticonvulsive effects in various preclinical models of seizures, as well as in patients with epilepsy. In this review, we provide an update on the pathologic role of NMDARs in epilepsy and an overview of the NMDAR antagonists that have been evaluated as anticonvulsive agents in clinical studies, as well as in preclinical seizure models.

## 1. Introduction

Epilepsy is a common neurological disorder, affecting about 1% of the general population, approximately 50 million people worldwide [1]. Despite the widespread use of anti-seizure medications (ASMs) over the past few decades in the management of epilepsy, about one-third of patients with epilepsy show no response to anti-seizure medication [2,3]. The International League Against Epilepsy (ILAE) defines drug-resistant epilepsy (DRE) as the “failure of adequate trials of 2 tolerated, appropriately chosen and used AED schedules (whether as monotherapies or in combination) to achieve sustained seizure freedom” [4]. The incidence of DRE was recently found to be about 19.6% of total epilepsy cases [5]. This calls for the need for further understanding of the mechanisms implicated in epileptogenesis, which could bolster treatment options.

Several pathways have been implicated to play a role in epileptogenesis [6], among which the desynchrony between neuronal excitation and inhibition is widely speculated to majorly contribute. Previous research on seizures from temporal lobe epilepsy (TLE) has demonstrated that glutamate levels rise in the extracellular fluid and that glutamate can directly activate *N*-methyl-D-aspartate receptors (NMDARs) and cause neuroexcitatory toxicity [7]. An overwhelming body of evidence exists centered around the role of NMDARs in several neurological disorders, including epilepsy. More recently, a sub-type of autoimmune encephalitis has been found to be associated with ~20% of epilepsy cases. This has further highlighted the role of the NMDAR complex in epileptogenesis [8]. These factors have formed the basis of research directed at both the preventive and therapeutic roles of NMDAR-modulating therapy in the management of neurological disorders.

The NMDAR complex is characterized by excitatory neurotransmitters (glutamate receptors) located on the synapses of interneurons regulating the balance between neuronal excitation and inhibition, and it has been implicated to play a role in epilepsy [9,10]. Several animal models have consistently shown that blocking NMDARs is effective in both the prevention and reversal of neurological disorders, including epilepsy [11]. A high sensitivity to modulation, as well as a proclivity for negative side effects, including neurotoxicity, has made it challenging for the development of newer agents. In this review, we highlight the role of NMDAR-guided therapy by providing an overview of the key pathophysiology, linking the role of the NMDAR complex in epilepsy with an emphasis on drugs currently in use, as well as on-going preclinical studies utilizing NMDAR-modulating therapy.

## 2. NMDAR Complex

The NMDAR is one of the ionotropic glutamate receptors (iGluR) serving as a target for action of the major excitatory neurotransmitter glutamate at the presynaptic terminal and post-synaptic membrane in the central nervous system (CNS) [12].

The NMDAR complex contributes to normal brain functioning, which begins in-utero. By providing neuronal excitation that promotes survival and efficient connectivity, NMDARs are involved in neurodevelopment [13]. This developmental period extends from the third trimester of pregnancy to the first several years of postnatal life in humans. NMDAR transmission is involved in the connectivity between hippocampal and prefrontal circuits [14]. NMDAR activation in hippocampal pathways controls an activity-dependent synaptic modification called long-term potentiation (LTP), contributing to learning and new memory formation [15,16].

In addition, the NMDAR complex is involved in spatial learning [17]. Furthermore, NMDAR transmission is involved in persistent neuronal firing, which has been implicated in underlying neurocognitive disorders and particularly in aversive mental states [18].

The receptor complex consists of a heterotetrametric structure in which an NR1 (GluN1) subunit is ubiquitous, and there are varying combinations of NR2 (GluN2) or NR3 (GluN3) subunits, with multiple binding sites for glutamate, polyamine, Mg^2+^, and glycine (Figure 1). The varying combinations of subunit binding sites determine the pharmacological regulation of the NMDAR. The glutamate binding site is situated on the NR2 subunit, and, similarly, the glycine binding site is situated on the NR1 subunit. These binding sites exhibit varying neuroanatomic expressions.

In the resting state, the channel is blocked by Mg^2+^ and remains equally permeable to Na^+^ and Ca^2+^ ions [19,20]. Membrane depolarization relieves the Mg^2+^ blockage, and the resulting neuronal excitation in-term mediates the NMDAR responses contributing to the neurotoxicity from excess Ca^2+^ ions. Neurotoxicity resulting from overexcitation has been widely speculated to play a major role in epilepsy [21].

### 2.1. NMDA Trafficking

NMDAR delivery to synapses and intracellular trafficking both depend on PDZ proteins [22]. NMDARs are not evenly distributed once they reach the neural surface, showing a higher concentration in postsynaptic densities and a lower one in extra-synaptic compartments [23]. Surface NMDARs are dynamically anchored in the postsynaptic density (PSD) region via an interaction between GluN2 subunits and proteins with PDZ-binding domains [24]. However, there are still questions about where receptor membrane trafficking occurs. The dysregulation of NMDAR trafficking has been implicated in several neuropsychiatric disorders in the past, and increasing evidence points to their involvement in epilepsy [25].

The trafficking of NMDARs to membranes was noted through an increase in synaptic and/or presynaptic NR1 subunits in a rat model of status epilepticus (SE) [26]. This increase in NMDA expression coincides with the loss of synaptic inhibitions through the internalization of (GABA)_A_ receptors implicated in the propagation of seizures to SE [27]. Furthermore, GRIN2A mutations can impact NMDAR trafficking overall by altering the levels of GLuN2A proteins and by altering GLuN2A membrane trafficking [28]. The defective interaction of protein binding sites involved in vesicular trafficking (SNX27) due to the phosphorylation of GluN2A has also been implicated in NMDAR trafficking defects [29]. In animal models and human epileptic brain tissue, G-protein-coupled receptor 40 (GPR40) affected N-methyl-D-aspartate (NMDA) receptor-mediated synaptic transmission through the regulation of NR2A and NR2B expressions on the surface of neurons [30]. Furthermore, the endocytosis of NMDARs and the binding of GPR40 with NR2A and NR2B were regulated through GPR40 [30]. Alterations in the interactions involved in NMDAR trafficking could open new avenues of therapeutic targets to alleviate neuronal overexcitation in epilepsy.

### 2.2. NMDA Modulation (Glycine and Other Sites)

The binding of a co-agonist at the glycine-mediated site (GMS) is necessary for NMDAR action, in addition to glutamate. The modulation of the GMS of the NMDAR is low, given the low saturation in vivo despite the high CSF concentrations of glycine [31]. D-serine serves as a major endogenous co-agonist of the NMDAR, and D-serine levels were recently found to be upregulated in intractable epilepsy [32,33,34]. It is possible that endogenous glycine does not fully stimulate NMDARs as suggested by the selective potentiation of the convulsant activity of NMDA by D-serine [35]. However, the therapeutic effect of glycine modulation is severely limited given the requirements of a high dose and a narrow therapeutic window and its severe adverse effects, such as oxidative damage, neurotoxicity, and nephrotoxicity [36].

When compared to other NMDAR subtypes, NR2B-containing receptors appear to contribute more favorably to pathogenic processes, such as epilepsy caused by excessive glutamatergic pathway activation. This makes them more of a preferential target for modulation [37]. A common mechanism involved in the allosteric modulation of NMDARs is through proton selectivity by shifting the pKa of the proton sensor [38,39]. This mechanism is involved in selective allosteric inhibition via ifenprodil, polyamines, and extracellular zinc at NR2A-containing receptors [40,41]. The allosteric modulation of NMDARs through a novel synthetic analogue of 24(S)-hydroxycholesterol-SGE-301 prevented the NMDAR dysfunction in patients with autoimmune encephalitis from NMDAR antibodies in cultured neurons [42]. A potential mechanism implicated was the prolonged decay time of NMDAR-dependent spontaneous excitatory postsynaptic currents suggesting a prolonged open time of the channel. More recently, miR-219, a microRNA, was implicated to play a regulatory role in the modulation of excitatory neurotransmission in epilepsy [43]. The upregulation of NR1 subunits was noted through an inverse relationship between miR-219 and NMDA-NR1 expression in the amygdala and hippocampus of patients with intractable mesial temporal lobe epilepsy [43].

### 2.3. NMDA mGluR and AMPA Interactions

Metabotropic glutamate receptors (mGluRs), which are a subtype of glutamate receptors, are members of G-protein-coupled receptors (GPCRs) involved in intracellular secondary messenger systems modulating neuronal excitability, which is of relevance in epilepsy [44]. mGluR5 responses have been found to be regulated by the activation of NMDARs via a protein kinase C (PKC) pathway [45,46]. Prior work has shown that mGluR5-positive modulators can attenuate the behavioral effects of NMDAR antagonists, PCP and MK-801 [47,48]. Despite the fact that there have not yet been any large clinical trials focusing on mGluR5 in epilepsy, selective group I mGluR antagonists were explored for their anticonvulsant effects in rodent models of epilepsy by Chapman et al., 2000, and Yan et al., 2005 [49,50]. However, the limitations behind their potential usefulness as anticonvulsant drugs would be due to the dominant effects of mGluR1 in cerebellar function and motor control [51]. Accordingly, patients who express autoantibodies against mGluR1 [52] or Homer-3 (a scaffolding protein for mGluRs) [53] exhibit signs of cerebellar dysfunction, such as ataxia.

The increased phosphorylation of NMDA and AMPA receptor subunits in rat models has been implicated in the regulation of synaptic plasticity and memory consolidation via the activation of ERK1/2 signaling [54]. In humans, Anti-GluA1 and Anti-GluA2 antibodies that target AMPAR subunits have been found in patients with epilepsy caused by autoimmune limbic encephalitis [55,56]. However, AMPAR autoantibodies were found to not have any interaction with NMDARs [57]. More recently, a combination of NMDAR and AMPAR antagonists in a mice model demonstrated that these receptor interactions could potentially contribute to delayed epileptogenesis through granule cell dispersion [58]. Though the response was a delay rather than the prevention of epileptogenesis, further clinical trials are warranted to study this interaction.

## 3. NMDAR Alterations and Their Role in Human Epilepsy

The alterations in NMDARs in epilepsy have been extensively investigated in the past through the use of a variety of techniques, such as gene expression, immunoblotting, and binding affinity techniques.

Through an in situ hybridization technique, Bayer et al., in 1995, showed that a loss of NR1-positive cells was associated with overall neuronal loss involving pyramidal cells [59]. Furthermore, NR2 subunit mRNA levels were increased in patients with hippocampal sclerosis (HS) [60]. In the dentate gyrus, there appears to be an increase in NR2 immunoreactivity that is associated with abnormal mossy fiber sprouting in this region [61]. It has been consistently demonstrated that the inhibition of the glutamate binding site (NR2 subunit) decreases granule cell hyperexcitability in cases showing mossy fiber sprouting in hippocampi [62,63,64]. More direct evidence from pyramidal neurons in human cortical slice preparations from patients with mesial temporal lobe epilepsy showed that an increased endogenous activity of NMDARs was associated with neuronal hyperexcitability [65]. More recently, in focal epilepsies, through the use of [(18)F]GE-179, a ligand that selectively binds to open NMDAR ion channels, McGinnity CJ et al. demonstrated NMDA channel overactivity through the use of positron emission tomography (PET) [66]. An alteration in NMDAR subunit (NR2B) composition in cortical dysplasia tissue has been shown to contribute to functional abnormalities due to decreased Mg^2+^ sensitivity of the receptor, which results in neuronal hyperexcitability [67]. In addition, increases in the levels of NR2B and 2D subunit mRNAs and functional NR2B-containing receptors (using a ligand-binding method) were noted in tuberous sclerosis [68]. Dysplastic neurons showed increased expressions of NR2B and 2C subunit mRNAs, whereas only NR2D mRNA was upregulated in giant cells, suggesting that dysplastic neurons and giant cells contribute differently to epileptogenesis in the tuberous sclerosis complex [68]. These studies emphasize that various alterations in the different subunits of NMDARs (especially NR1 and NR2 subunits) in different brain regions could be responsible for seizure development accordingly among the several types of epilepsies.

In human patients with symptomatic epilepsies, the regulation of the GluN2B subunit of the NMDAR complex via NRG1-ErbB4-Src signaling pathways was identified as a potential modulating target through the use of the immunoblotting technique [69]. In patients with intractable temporal lobe epilepsies, D-serine and NMDAR1 expressions were significantly increased [34]. These observations highlight the importance of neurochemical targeting, which can be further explored in the future to guide anti-NMDAR complex therapies.

Moreover, a variety of animals models of seizures and epilepsy have demonstrated alterations in NMDAR expressions and protein levels, although the results vary depending on the NMDAR subunits, brain regions, and animal species assessed. This has been comprehensively reviewed in the literature [33,69,70,71].

### 3.1. Genetic Mutations of the NDMA Receptor

Various genetic expressions of the subunits of the NMDAR (GluN1, GluN2A-2D, and GluN3A-3B) can contribute to the development of distinct clinical phenotypes [72,73]. The genetic mutations in patients with epilepsy are classified broadly as loss-of-function, no-change, and gain-of-function mutations [73].

The GluN1 subunit, encoded by GRIN1, is typically involved in loss-of-function mutations contributing to structural changes resulting in a wide range of epilepsies of variable semiology (spasms, tonic and atonic seizures, hypermotor seizures, focal dyscognitive seizures, febrile seizures, generalized seizures, status epilepticus, myoclonic seizures, etc.) [74]. Reportedly, up to half of GRIN1 mutations are loss-of-function mutations, with the rest being gain-of-function mutations. Hence, the co-existence of both hypo-functioning and hyper-functioning NMDARs within the same disease phenotype invariably contributes to electrophysiological imbalance [10,12]. However, this relation of NMDAR function to the pathogenesis of epilepsy still needs further clarification. GRIN2A mutations, which can alter the GluN2A receptor, occur more so than any other of the NMDAR subunit mutations [75]. About 70% of GRIN2A variants are likely to lead to the development of epilepsy, whereas 30% of individuals with GRIN2B variants have epilepsy [76,77]. GluN2 subunit mutations may control epileptiform abnormalities arising from the hippocampus [78]. GRIN2D mutations are associated with treatment refractory epileptic encephalopathy [79].

Various alterations in subunits in different brain regions are, thus, held accountable for seizure development in different types of epilepsies. Identifying receptor mutations has, hence, been implicated to contribute to a personalized medicine approach in epilepsy treatment. Modeling NMDAR dysfunction with neurons derived from human induced pluripotent stem cells (iPSCs), as well as identifying signaling pathways, has been suggested to further contribute to the development of drugs targeting the NMDAR complex of gene regulatory variants [80].

### 3.2. Anti-NMDAR Encephalitis

Autoimmune encephalitis from anti-NMDAR antibodies has gained significant attention over the past decade [81,82] to the extent of being regarded as its own entity of epilepsy [83,84]. Anti-NMDA-NR2A/B antibodies are present in ~20% of patients with epilepsy [8]. The autoantibodies against NMDARs can cause a reversible loss of NMDAR function on the surface of neurons [85]. Several mechanisms are involved:(a)Internalization of NMDAR,(b)Disruption of interaction of NMDAR with EphB2R,(c)Decreased long-term potential (LTP), which can lower the threshold for seizures [86,87,88,89].

Reductions in excitatory neurotransmission caused by NMDAR antibodies in an vitro rat model may play a role in NMDA encephalitis [90]. These mechanisms invariably result in a state of excitatory and inhibitory chemical imbalance contributing to seizures in a significant number of patients. These are typically improved by the prompt use of immunomodulatory therapy [91]. A marked decrease in anti-NMDA-NR1 antibody titers was seen following the administration of steroids, intravenous immunoglobulin G (IVIG), or plasmapheresis/plasma exchange, which are part of the first-line management strategies, and following the administration of steroid-sparing agents: rituximab, cyclophosphamide, or both [85,92,93].

## 4. NMDAR Modulators Currently in Use

### 4.1. Ketamine

Ketamine acts as a noncompetitive NMDA antagonist and has low affinity for the NMDAR at the phencyclidine site within the ionotropic channel [94]. It was first synthesized in 1962. Conventionally used as an anesthetic agent due to its rapid onset of action and short half-life, ketamine induces a state of “dissociative anesthesia” resulting from an overall CNS depressant effect rather than an inhibitory effect [66,95]. The anticonvulsant effects of ketamine can augment the seizure protective effect of benzodiazepine loading [96,97]. This therapeutic benefit may result from the upregulation of NMDARs during prolonged seizures at the same time when there is an overall decrease in sensitivity to GABA agonists [98,99,100].

Animal models have consistently shown the synergistic anticonvulsant effects of ketamine therapy in combination with other antiepileptic medications demonstrating both dose- and time-dependent adjuvant effects [101,102]. A multicenter retrospective study provided preliminary data on the safety and efficacy of intravenous ketamine use in the treatment of refractory status epilepticus, which is defined as seizure activity that does not respond to two antiepileptic drugs at appropriate doses [103]. High-dose and the early initiation of ketamine infusions (2.2 mg/kg/h) were associated with a decrease in seizure burden in patients with super-refractory status epilepticus (SRSE) [104]. The favorable effect on the hemodynamic profile, which is due to ketamine’s sympathomimetic properties, unique benefits of conscious sedation, and overall efficacy in treating refractory seizures, makes ketamine an agent to consider using in the management of patients with severe acute traumatic brain injury [105,106,107]. In critically ill patients where polypharmacy is a concern, ketamine use can limit the need for the further escalation of sedation and thereby help to avoid the need for intubation [108,109]. Further prospective randomized control trials are required to establish a consensus statement on dosing ketamine in refractory epilepsy management in adult and pediatric populations.

The adverse effects of ketamine infusions include neuropsychiatric symptoms, such as hallucinations and delirium, and arrhythmias. There has been a rare report of new-onset seizures in a pediatric patient [100]. Ketamine has been found to induce dose-dependent neuronal injury in animal models through apoptosis, particularly in the frontal cortex and hippocampal regions [110,111]. This, at the same time, is offset by the neuroprotective effect conferred by ketamine through an increase in regional cerebral blood flow volumes, thereby limiting the damage from an incomplete cerebral ischemia [112,113]. More recently, the therapeutic effect of ketamine has been explored in targeted therapy on spreading depolarizations, which represent a unique pathophysiology contributing to secondary injury progression in severe acute brain injury [114,115,116]. Furthermore, an interplay between seizures and spreading depolarizations has been suggested [116,117]. This further emphasizes the fact that NMDARs are involved in this epiphenomenon of secondary brain injury progression. The prevention of secondary brain injury spread has potential to improve outcomes in critically ill patients with sub-arachnoid hemorrhage, traumatic brain injury, and malignant ischemic strokes.

### 4.2. Memantine

Memantine is a noncompetitive, open-channel NMDAR antagonist that blocks NMDAR ion channels by binding to or near Mg^2+^ binding sites preferentially when the receptor channel is open, thereby inhibiting the prolonged influx of Ca^2+^ ions with near-normal physiological NMDA activity [118]. The efficacy of memantine as an anticonvulsant for both monotherapy and in combination with other AEDs was explored in preclinical work in the early 1980s [119,120,121].

In a lithium–pilocarpine mice model, memantine prevented cognitive impairments post-status epilepticus [122]. In a pentylenetetrazole (PTZ) model of seizures, memantine prevented convulsions and the development of morphological changes [123]. Memantine had a significant neuroprotective effect on hippocampal and cortical neurons in culture against glutamate and NMDA excitotoxicity [124]. A mice model showed evidence of a decrease in the frequency of induced seizures with the addition of memantine [125]. However, the addition of memantine to the AED regimen in GRIN2B-mutation-related encephalopathy did not result in any significant decrease in the frequency of seizures in a group of six patients [126].

Given the paucity of clinical data, more clinical trials testing the anticonvulsant effects of memantine as an add-on therapy on seizure types in patients with epilepsy are needed.

### 4.3. Amantadine

Amantadine was originally used in the management of influenza. It acts by increasing the release and inhibiting the reuptake of dopamine in the brain. In addition to the several other pharmacological actions of amantadine, its role in NMDAR blockage by increasing the rate of channel closure was demonstrated in the early 1990s [127].

The safety and efficacy of amantadine as an add-on therapy in pediatric refractory generalized seizures was first evaluated in a small case series of four patients by Shahar et al. in 1993 [128,129]. In a larger retrospective review of 13 patients, a target amantadine dose of 4 to 7 mg/kg/day was utilized in conjunction with other AEDs. A total of 58% of patients had at least a 50% seizure reduction, with the majority (nearly 86%) of responders sustaining a seizure reduction greater than 90% at 12 months following treatment initiation [130]. Of note, no renal, hepatic, or hematologic toxicity was noted in this study, and other adverse effects included vomiting, behavioral changes, headache, dizziness, and weight loss.

Its favorable pharmacological profile, including the potential improved effects on cognitive recovery [131,132,133], make amantadine a promising agent of choice to explore in critically ill patients with refractory epilepsy. Further larger, multicenter trials are needed to study the efficacy of amantadine add-on therapy in refractory epilepsy.

### 4.4. Magnesium Sulphate

The potential for convulsions to occur in states of Mg^2+^ depletion prompted some researchers to investigate the metabolism of this metal in epilepsies [134]. Experiments have shown that Mg^2+^ blocks Ca^2+^ within the NMDAR channel, which, in turn, is relieved by cellular depolarization [135]. The anticonvulsant effects of the systemic administration of Mg^2+^ have been studied in animal models [136]. An increase in brain Mg^2+^ concentrations in rat brains has been found to be associated with an increased seizure threshold and resistance to NMDA-stimulated hippocampal seizures [137]. The inhibition of NMDARs is central to Mg^2+^ anticonvulsant effects [138].

In two patients with juvenile-onset Alpers’ syndrome, intravenous Mg^2+^ treatment reduced refractory epilepsy with recurring status epilepticus and bouts of epilepsia partialis continua [139]. Furthermore, in infants with infantile spasms, the addition of magnesium sulphate to adrenocorticotropic hormone (ACTH) showed significantly improved durations of seizure-free periods in an open-label, randomized, controlled study [140]. In humas, systemic administrations of magnesium sulphate are the standard of care in the management of eclampsia in pregnancy [141]. Concerns remain over the possibilities of hypermagnesemia toxicity in eclampsia treatment [142].

### 4.5. Felbamate

Felbamate is a propanediol dicarbamate derivative that was first approved as an ASM by the U.S. Food and Drug Administration (FDA) in 1993 [143]. Several mechanisms, including the blockage of voltage-gated sodium channels and dual actions on both excitatory (NMDA) and inhibitory (GABA) mechanisms, have been postulated to contribute to the anticonvulsant effect of felbamate [144].

In addition, the specific selectivity of felbamate to the NR2B subunit of NMDA has been noted. NR2B mutations have been implicated to play a role in epileptogenesis in mouse models [145]. Felbamate’s efficacy as add-on therapy is well-established for intractable partial seizures, infantile spasms, and Lennox-Gastaut syndrome [146,147]. Serious adverse effects of aplastic anemia and liver failure have been reported, which has limited the use of felbamate as add-on therapy in partial epilepsy [148].

### 4.6. Remacemide

Remacemide is a noncompetitive, low-affinity, NMDAR antagonist. A rapid and reversible inhibition of the NMDA current was first observed in rat forebrain membranes [149]. Its anticonvulsant effects in animal models of epilepsy were observed in a dose range of 6–60 mg/kg [150]. Remacemide hydrochloride was shown to have therapeutic activity in patients with medically refractive epilepsy at a dose of 600 mg/day [151]. Chadwick et al. [152] and Jones et al. [153] evaluated the role of the remacemide QID regimen as an add-on in medically refractory epilepsy.

It is worth noting that in patients with newly diagnosed epilepsy, when compared to other AEDs, such as carbamazepine, remacemide had no benefit as monotherapy [154]. A double-blind, parallel-group trial comparing remacemide with carbamazepine in partial or generalized tonic–clonic seizures showed a better cognitive and psychomotor profile at the cost of inferior seizure recurrence with the use of remacemide [155]. Some common adverse effects observed are dizziness, somnolence, and gastrointestinal symptoms. Diplopia and fatigue have been observed when used as adjunctive therapy with conventional AEDs [156].

### 4.7. Riluzole

2-Amino-6trifluoromethoxy benzothiazole (riluzole) was first noted to have anticonvulsant effects in the 1980s [157]. Several mechanisms involving glutamate modulation were noted. The presynaptic release of glutamate was reduced by the inhibition of voltage-gated Na^+^ currents in hippocampal neurons [158]. Riluzole can prevent Ca^2+^ entry via the NMDA channel, thereby blocking NMDAR activation [159]. A modulatory effect of riluzole on glutamate clearance has also been noted on the glutamate transporters expressed on neurons and glia [160].

Preclinical data have shown anticonvulsant effects in a variety of seizure models; in the rat dentate gyrus of both pilocarpine- and GBL-induced seizure models, riluzole was found to be more effective in reducing seizure activity than VPA [161]. Riluzole reduced seizure duration in a rat electroconvulsive shock model of epilepsy [162]. Furthermore, the inhibition of spontaneous glutamine transport by riluzole was demonstrated in hippocampal neurons [163]. In a CA3 in vitro rat slice model, the sodium channel blockage of riluzole resulted in decreased hippocampal epileptiform activity [164]. Despite the evidence from animal models, the effects of riluzole in human patients with epilepsy are yet to be evaluated in any clinical study.

### 4.8. Dizocilpine or MK-801

MK-801 is a noncompetitive NMDA antagonist that was first demonstrated to have antiepileptic effects in a model of induced seizures through the low-frequency kindling technique by Minabe et al. in 1992 [165]. MK-801 is a special NMDAR antagonist due to its well-demonstrated effects in both use-dependent and voltage-dependent manners via the blockade of ion permeation [166,167,168,169]. A large number of preclinical investigations have demonstrated its anticonvulsant properties in several models, including seizures induced by NMDA, quinolinic acid, lindane, 4-aminopyridine, caffeine, picrotoxin, bicuculline, cocaine, kainic acid, strychnine, pentylenetetrazole (PTZ), or hyperbaric oxygen in rodents [170,171,172,173,174,175,176,177,178,179,180,181]. However, more recent evidence based on the stargazer mice trial suggests that MK-801 has a paradoxical pro-seizure effect [182].

In tetramethylenedisulfotetramine (TMDT)-induced tonic–clonic seizures, the combination of diazepam and MK-801 had a synergistic anticonvulsant effect [183]. Status epilepticus was aborted, and mortality was eliminated with the combination of diazepam and dizocilpine in a rat model of SE induced by very high doses of lithium and pilocarpine [178], as well as in a model of soman-induced SE [184,185]. These observations suggest the possibility of clinical benefits of combinations of MK-801 and other anti-seizure medications. Due to concerns regarding the schizophrenia-like behaviors frequently noted in animal models, MK-801 has not been explored in human subjects [186].

### 4.9. Dextromethorphan

Initially introduced as an anti-tussive agent, owing to its several mechanisms of actions, dextromethorphan has found potential use as both an analgesic and anticonvulsive agent [187,188,189]. Dextromethorphan was noted to have noncompetitive NMDAR blocking effects with an efficacy similar to that of controlled substances, such as phencyclidine and ketamine, at high doses.

Dextromethrophan, added to existing AEDs at doses of 40 mg and 50 mg every 6 h (160 and 200 mg/day, respectively) for 8 weeks, resulted in significant improvements in seizure control in patients with drug-resistant, localization-related epilepsies [189]. Furthermore, a randomized, open-label trial of dextromethorphan in Rett’s syndrome showed evidence of significant dose-dependent improvements in clinical seizures [190]. This preliminary evidence needs further validation in larger cohorts.

The adverse effects of dextromethorphan include nystagmus, slurred speech, light-headedness, and fatigue at high doses [191]. In addition, sudden tonic–clonic movements and confusion have been reported to be caused by the toxicity associated with dextromethorphan in a case report [192].

### 4.10. Ifenprodil

Ifenprodil (4-[2-(4-benzylpiperidin-1-yl)-1-hydroxypropyl]phenol) is a selective antagonist of the GluN2B subtype of NMDARs [193]. The anticonvulsant effect of ifenprodil has been investigated in several animal models of epilepsy, including induced seizures by NMDA, spermine, lindane, and PTZ in rodents [194,195,196,197,198]. Furthermore, both age-dependent and activation-dependent anticonvulsant actions of ifenprodil were noted [199]. In five human patients with refractory epilepsy caused by malformations of cortical development (MCD), ifenprodil had specific antiepileptic effects by reducing pyramidal cell neural excitability [200]. More recently, in temporal lobe epilepsy medicated by the GluN2B subtype of NMDARs, intraperitoneal ifenprodil administration (20 mg/kg) resulted in the suppression of a number of chronic seizures and an anti-ictogenic effect [201].

When used in combination with other AEDs, the threshold for seizures was increased without influencing the anticonvulsant actions of other drugs (carbamazepine, diphenylhydantoin, phenobarbital, and valproate) [202].

## 5. Preclinical Studies with Newer NMDAR Modulators

A variety of NMDR modulators have been investigated in preclinical studies using various seizure models [11]. More recent studies have also tested newer agents that modulate NMDARs (Table 1). In a mouse model on clonic seizures induced by PTC, the involvement of the NMDAR pathway in the anticonvulsant effect of licofelone (dual 5-lipoxygenase/cyclooxygenase inhibitor) was demonstrated by combining the noncompetitive NMDAR antagonist MK-801 with licofelone (5 mg/kg) [203]. In addition, a lower dose of licofelone did not have an anticonvulsant effect [203]. Recently, the use of GNE-0723 (a positive allosteric modulator of GluN2A-subunit-containing NMDARs) reduced low-frequency oscillatory and epileptiform activities in J20 mice [204]. The GluN2B-selective antagonist Ro 25-6981 suppressed the tonic phase of generalized tonic–clonic seizures in a PTZ model of infantile rats [205].

## 6. Adverse Effects of NMDAR Antagonist in Clinical Settings

The inhibition of the major excitatory neurotransmitter glutamate is bound to have adverse effects that can potentially limit its potential for clinical application. In a clinical investigation, the competitive NMDAR antagonist D-CPP-ene increased seizures in three out of eight patients with epilepsy, raising the possibility that a sudden decline in NMDAR function could lead to an imbalance between the excitatory and inhibitory systems [207]. A selective blockade of NMDARs without affecting normal function remains a necessity for acceptability in clinical practice. An ideal agent should hence serve the role of an “uncompetitive” antagonist by relying on prior receptor activation by the agonist [208].

Adverse effects, such as hallucinations, lightheadedness, dizziness, fatigue, headaches, out-of-body sensation, and sensory changes, have been reported with NMDA antagonists. Early memory impairments and schizophrenia-like symptoms have been linked to NMDAR hypofunction with the use of antagonists [209]. An early blockade of NMDARs in rat brains has been found to trigger apoptotic neurodegeneration [210]. Brain growth, long-term potentiation, neuronal migration, and synaptic pruning are all significantly influenced by NMDAR activity [211]. Accordingly, the in utero use of NMDA antagonists may disrupt brain development by contributing directly to the disconnection of circuits between the hippocampus and frontal cortex [212].

Impairments in learning ability and memory as the result of NMDAR antagonist therapy, especially in early life, have been extensively studied in rat models [213,214]. These were notably more evident upon direct NMDAR antagonist injection into the amygdala and hippocampus [215]. In human subjects, following ketamine infusions, disruptions in frontal and hippocampal responses contributing to working memory were demonstrated through the use of fMRI imaging [216]. However, it is worth noting that NMDAR blockade impaired learning consolidation without having any effect on the memory retrieval of previously learned tasks [217,218].

Furthermore, in addition to intravenous administration, as extensively previously discussed, other routes of administration can affect the appearance of specific cytotoxic side effects. For example, spinal cord pathology has been noted with intrathecal administration [219]. The intrathecal administration of ketamine in a therapeutically appropriate concentration and dosage had a deleterious effect on rabbits’ central nervous systems [220]. In dogs, the subarachnoid administration of ketamine was found to be associated with histological spinal cord alterations, including gliosis, axonal edema, central chromatolysis, lymphocyte infiltration, and fibrous thickening of the dura mater [221]. Additionally, a terminally ill patient with cancer who had continuous intrathecal ketamine infusions at a rate of 5 mg/day for a period of 3 weeks was reported by Karpinski et al. to have post-mortem CNS histological abnormalities of subpial spinal cord vacuolation [222].

## 7. Conclusions and Perspectives

Overwhelming evidence now indicates that the NMDAR complex plays a critical role in seizure disorders. This comes from both preclinical and clinical studies that have found alterations in NMDAR expression and function in seizure models, as well as in patients with epilepsy. Accordingly, various NMDAR modulators have been tested in various animal models of seizures, and in these studies, they have shown efficacy in suppressing different types of seizures. Although few NMDAR antagonists have been evaluated in clinical trials in epileptic patients, the results are promising overall and have opened a new avenue for the treatment of epilepsy. However, long-term therapy with a newer class of NMDAR antagonists warrants further longitudinal studies, especially to assess their safety in this patient population.

## Figures and Tables

**Figure 1 pharmaceuticals-15-01297-f001:**
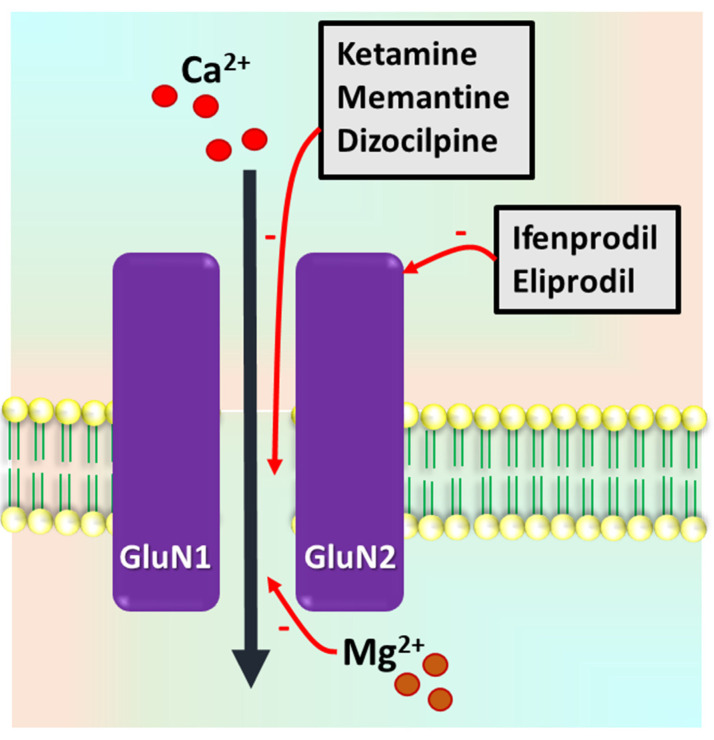
Schematic representation of *N*-methyl-D-aspartate receptor (NMDAR) complex and binding sites for GluN1 and GluN2 for some NMDAR antagonists. Ifenprodil and eliprodil mainly bind to GluN2B subunit. Mg^2+^, dizocilpine (MK-801), ketamine, and memantine act as noncompetitive antagonists with binding sites inside the ion channel pore region.

**Table 1 pharmaceuticals-15-01297-t001:** Anticonvulsive effects of newer NMDAR antagonists in pre-clinical studies.

Substance	Effect on NMDARs	Seizure Model	Effect	Ref.
GNE-0723	Positive allosteric modulator of GluN2A	Mouse model of Dravet syndrome	↓ Low-frequency oscillatory and epileptiform activities	[204]
Ro 25-6981	Selective GluN2B antagonist	PTZ model in infantile (12-day-old, P12) and juvenile (25-day-old, P25) rats	↓ PTZ-induced seizures in infantile, but not juvenile, rats	[205]
PEAQX	Selective GluN2A antagonist	PTZ-induced generalized seizures	Age-dependent differences in anticonvulsant effects in PTZ-induced seizures and epilepsy after discharge	[206]
DDBM	Both GluN1 and GluN2 antagonist	Rat ECS model of epilepsy	↓ Seizure behaviors in rats	[162]

DDBM, l indolyl, [2-(1,1-Dimethyl-1,3-dihydro-benzo[e]indol-2-ylidene)-malonaldehyde]; ECS, electroconvulsive shock; PTZ, pentylenetetrazole.

## Data Availability

Data sharing not applicable.

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
