# Peer review of "Targeting NMDA Receptor Complex in Management of Epilepsy"

_pharmaceuticals, 2022, doi:10.3390/ph15101297_

Round 1
Reviewer 1 Report
This review attempts to showcase NMDA in epilepsy. However it has many short-comings that need to be addressed before it can have an impact. Overall, the discussion of NMDA is lacking in many aspects. For example, NMDA trafficking, NMDA modulation (ie. Glycine and other sites), NMDA mGluR and AMPA interactions. It is possible that there is no evidence of such interactions, yet their possible role in human epilepsy needs to be addressed.
Author Response
Reviewer’s Comments
Comment. This review attempts to showcase NMDA in epilepsy. However it has many short-comings that need to be addressed before it can have an impact. Overall, the discussion of NMDA is lacking in many aspects. For example, NMDA trafficking, NMDA modulation (ie. Glycine and other sites), NMDA mGluR and AMPA interactions. It is possible that there is no evidence of such interactions, yet their possible role in human epilepsy needs to be addressed.
Response.
We appreciate your feedback and review of our manuscript. We have responded to each critique and revised the manuscript accordingly. As a result, the revised manuscript is stronger. Please find attached below the topics now included in the manuscript.
“2.1 NMDA trafficking
NMDAR delivery to synapses and intracellular trafficking both depend on PDZ proteins [1]. NMDARs are not evenly distributed once they reach the neural surface, showing a higher concentration in postsynaptic densities and a lower one in extrasynaptic compartments [2]. Surface NMDARs are dynamically anchored in the postsynaptic density (PSD) region via an interaction between GluN2 subunits and proteins with PDZ-binding domains [3]. Although there are still questions about where the receptor membrane trafficking occurs. Dysregulation of NMDAR trafficking has been implicated in several neuropsychiatric disorders in the past, and increasing evidence points to their involvement in epilepsy [4].
Trafficking of NMDA receptors to membranes were noted through increase in synaptic and/or presynaptic NR1 subunits in a rat model of status epilepticus (SE)[5]. This increase in NMDA expression coincides with the loss of synaptic inhibitions (through internalization of (GABA)A receptor implicated in propagation of seizure to SE[6]. Furthermore, GRIN2A mutations can overall impact NMDAR trafficking through altering levels of GLuN2A protein, and alteration of GLuN2A membrane trafficking[7]. Defective interaction of protein binding sites involved in vesicular trafficking (SNX27) due to phosphorylation of GluN2A were also implicated in NMDAR trafficking defects[8]. In animal models and human epileptic brain tissue, G protein–coupled receptor 40 (GPR40) affected N-methyl-D-aspartate (NMDA) receptor–mediated synaptic transmission through regulation of NR2A and NR2B expression on the surface of neurons[9]. Furthermore, endocytosis of NMDA receptors and binding of GPR40 with NR2A and NR2B were regulated through GPR40[9]. Alterations in interactions involved in NMDAR trafficking could open new avenues of therapeutic targets to alleviate neuronal overexcitation in epilepsy.
2.2 NMDA modulation (Glycine and other sites)
The binding of a co-agonist at the glycine mediated site (GMS) is necessary for NMDAR action in addition to glutamate. Modulation of GMS of the NMDAR is low given low-saturation in-vivo despite high CSF concentrations of glycine [10]. D-serine serves as a major endogenous co-agonist of the NMDAR, and expression of D-serine was recently found to be upregulated in intractable epilepsy [11-13]. It is possible that endogenous glycine does not fully stimulate NMDA receptors as suggested by the selective potentiation of the convulsant activity of NMDA by D-serine [14]. However, the therapeutic effect of glycine modulation is severely limited given requirements for high dose, narrow therapeutic window and severe adverse effects such as oxidative damage, neurotoxicity, and nephrotoxicity[15].
When compared to other NMDAR subtypes, NR2B-containing receptors appear to contribute more favorably to pathogenic processes such as epilepsy connected to excessive glutamatergic pathway activation. This makes it more of a preferential target to modulation [16]. A common mechanism involved in allosteric modulation of NMDAR is through proton selectivity by shifting pKa of the proton sensor [17,18]. This mechanism is involved in selective allosteric inhibition by ifenprodil, polyamines and extracellular zinc at NR2A-containing receptor [19,20]. Allosteric modulation of NMDAR through novel synthetic analogue of 24(S)-hydroxycholesterol
- SGE-301 prevented the NMDAR dysfunction caused by patients with autoimmune encephalitis from NMDAR antibodies in cultured neurons [21]. A potential mechanism implicated was through prolonged decay time of NMDAR-dependent spontaneous excitatory postsynaptic currents suggesting a prolonged open time of the channel. More recently, miR-219- a microRNA was implicated in a regulatory role in modulation of excitatory neurotransmission in epilepsy [22]. Upregulation of NR1 sub-units were noted through an inverse relationship between miR-219 and NMDA-NR1 expression in amygdala and hippocampus of patients with intractable mesial temporal lobe epilepsy [22].
2.3 NMDA mGluR and AMPA interactions
Metabotropic glutamate receptors (mGluRs) which are a subtype of glutamate receptors, are members of G-protein-coupled receptor (GPCR) involved in intracellular secondary messenger systems modulating neuronal excitability which is of relevance in epilepsy[23]. mGluR5 responses have been found to be regulated through activation of NMDA receptors by a protein kinase C (PKC) pathway [24,25]. Prior work has shown that mGluR5 positive modulators can attenuate behavioral effects of NMDA receptor antagonists, PCP and MK-801[26,27]. Despite this fact that there have not yet been any large clinical trials focusing on mGluR5 in epilepsy. Selective group I mGluR antagonists were explored for their anti-convulsant effects in rodent models of epilepsy by Chapman et al., 2000; and Yan et al., 2005 [28,29]. However, limitation behind their potential usefulness as an anticonvulsant would be due to dominant effects of mGluR1 in cerebellar function and motor control [30].
Increased phosphorylation of NMDA and AMPA receptor subunits in rat models have been implicated in regulation of synaptic plasticity and memory consolidation through activation of ERK1/2 signaling[31]. In humans, Anti-GluA1 and Anti-GluA2 antibodies that target AMPAR subunits have been found in patients with epilepsy from autoimmune limbic encephalitis[32,33]. However, AMPAR autoantibodies did not have any interaction with NMDAR[34]. More recently, a combination of NMDAR and AMPAR antagonist in a mice model demonstrated that these receptor interactions could potentially contribute to delayed epileptogenesis through granule cell dispersion[35]. Though the response was a delay rather than prevention of epileptogenesis, further clinical trials are warranted to study this interaction.”
References
- Kim, E.; Sheng, M. PDZ domain proteins of synapses. Nature Reviews Neuroscience 2004, 5, 771-781, doi:10.1038/nrn1517.
- Ladépêche, L.; Dupuis, J.P.; Groc, L. Surface trafficking of NMDA receptors: Gathering from a partner to another. Seminars in Cell & Developmental Biology 2014, 27, 3-13, doi:https://doi.org/10.1016/j.semcdb.2013.10.005.
- Bard, L.; Sainlos, M.; Bouchet, D.; Cousins, S.; Mikasova, L.; Breillat, C.; Stephenson, F.A.; Imperiali, B.; Choquet, D.; Groc, L. Dynamic and specific interaction between synaptic NR2-NMDA receptor and PDZ proteins. Proceedings of the National Academy of Sciences 2010, 107, 19561-19566, doi:doi:10.1073/pnas.1002690107.
- Lau, C.G.; Zukin, R.S. NMDA receptor trafficking in synaptic plasticity and neuropsychiatric disorders. Nat Rev Neurosci 2007, 8, 413-426, doi:10.1038/nrn2153.
- Wasterlain, C.G.; Naylor, D.E.; Liu, H.; Niquet, J.; Baldwin, R. Trafficking of NMDA receptors during status epilepticus: therapeutic implications. Epilepsia 2013, 54 Suppl 6, 78-80, doi:10.1111/epi.12285.
- Mele, M.; Costa, R.O.; Duarte, C.B. Alterations in GABA(A)-Receptor Trafficking and Synaptic Dysfunction in Brain Disorders. Front Cell Neurosci 2019, 13, 77, doi:10.3389/fncel.2019.00077.
- Addis, L.; Virdee, J.K.; Vidler, L.R.; Collier, D.A.; Pal, D.K.; Ursu, D. Epilepsy-associated GRIN2A mutations reduce NMDA receptor trafficking and agonist potency – molecular profiling and functional rescue. Scientific Reports 2017, 7, 66, doi:10.1038/s41598-017-00115-w.
- Mota Vieira, M.; Nguyen, T.A.; Wu, K.; Badger, J.D.; Collins, B.M.; Anggono, V.; Lu, W.; Roche, K.W. An Epilepsy-Associated GRIN2A Rare Variant Disrupts CaMKIIα Phosphorylation of GluN2A and NMDA Receptor Trafficking. Cell Reports 2020, 32, 108104, doi:https://doi.org/10.1016/j.celrep.2020.108104.
- Yang, Y.; Tian, X.; Xu, D.; Zheng, F.; Lu, X.; Zhang, Y.; Ma, Y.; Li, Y.; Xu, X.; Zhu, B.; et al. GPR40 modulates epileptic seizure and NMDA receptor function. Science Advances 2018, 4, eaau2357, doi:doi:10.1126/sciadv.aau2357.
- Bergeron, R.; Meyer, T.M.; Coyle, J.T.; Greene, R.W. Modulation of N-methyl-D-aspartate receptor function by glycine transport. Proc Natl Acad Sci U S A 1998, 95, 15730-15734, doi:10.1073/pnas.95.26.15730.
- Mothet, J.P.; Le Bail, M.; Billard, J.M. Time and space profiling of NMDA receptor co-agonist functions. J Neurochem 2015, 135, 210-225, doi:10.1111/jnc.13204.
- Zhu, S.; Paoletti, P. Allosteric modulators of NMDA receptors: multiple sites and mechanisms. Curr Opin Pharmacol 2015, 20, 14-23, doi:10.1016/j.coph.2014.10.009.
- Zhang, X.; Hu, B.; Lu, L.; Xu, D.; Sun, L.; Lin, W. D-serine and NMDA Receptor 1 Expression in Patients with Intractable Epilepsy. Turk Neurosurg 2021, 31, 76-82, doi:10.5137/1019-5149.Jtn.28138-19.2.
- Singh, L.; Oles, R.J.; Tricklebank, M.D. Modulation of seizure susceptibility in the mouse by the strychnine-insensitive glycine recognition site of the NMDA receptor/ion channel complex. Br J Pharmacol 1990, 99, 285-288, doi:10.1111/j.1476-5381.1990.tb14695.x.
- Meftah, A.; Hasegawa, H.; Kantrowitz, J.T. D-Serine: A Cross Species Review of Safety. Front Psychiatry 2021, 12, 726365, doi:10.3389/fpsyt.2021.726365.
- Mony, L.; Kew, J.N.; Gunthorpe, M.J.; Paoletti, P. Allosteric modulators of NR2B-containing NMDA receptors: molecular mechanisms and therapeutic potential. Br J Pharmacol 2009, 157, 1301-1317, doi:10.1111/j.1476-5381.2009.00304.x.
- Zhang, J.-B.; Chang, S.; Xu, P.; Miao, M.; Wu, H.; Zhang, Y.; Zhang, T.; Wang, H.; Zhang, J.; Xie, C.; et al. Structural Basis of the Proton Sensitivity of Human GluN1-GluN2A NMDA Receptors. Cell Reports 2018, 25, 3582-3590.e3584, doi:https://doi.org/10.1016/j.celrep.2018.11.071.
- Regan, M.C.; Zhu, Z.; Yuan, H.; Myers, S.J.; Menaldino, D.S.; Tahirovic, Y.A.; Liotta, D.C.; Traynelis, S.F.; Furukawa, H. Structural elements of a pH-sensitive inhibitor binding site in NMDA receptors. Nature Communications 2019, 10, 321, doi:10.1038/s41467-019-08291-1.
- Schreiber, J.A.; Schepmann, D.; Frehland, B.; Thum, S.; Datunashvili, M.; Budde, T.; Hollmann, M.; Strutz-Seebohm, N.; Wünsch, B.; Seebohm, G. A common mechanism allows selective targeting of GluN2B subunit-containing N-methyl-D-aspartate receptors. Communications Biology 2019, 2, 420, doi:10.1038/s42003-019-0645-6.
- Low, C.M.; Zheng, F.; Lyuboslavsky, P.; Traynelis, S.F. Molecular determinants of coordinated proton and zinc inhibition of N-methyl-D-aspartate NR1/NR2A receptors. Proc Natl Acad Sci U S A 2000, 97, 11062-11067, doi:10.1073/pnas.180307497.
- Mannara, F.; Radosevic, M.; Planagumà, J.; Soto, D.; Aguilar, E.; García-Serra, A.; Maudes, E.; Pedreño, M.; Paul, S.; Doherty, J.; et al. Allosteric modulation of NMDA receptors prevents the antibody effects of patients with anti-NMDAR encephalitis. Brain 2020, 143, 2709-2720, doi:10.1093/brain/awaa195.
- Hamamoto, O.; Tirapelli, D.P.d.C.; Lizarte Neto, F.S.; Freitas-Lima, P.; Saggioro, F.P.; Cirino, M.L.d.A.; Assirati Jr, J.A.; Serafini, L.N.; Velasco, T.R.; Sakamoto, A.C.; et al. Modulation of NMDA receptor by miR-219 in the amygdala and hippocampus of patients with mesial temporal lobe epilepsy. Journal of Clinical Neuroscience 2020, 74, 180-186, doi:https://doi.org/10.1016/j.jocn.2020.02.024.
- Ure, J.; Baudry, M.; Perassolo, M. Metabotropic glutamate receptors and epilepsy. J Neurol Sci 2006, 247, 1-9, doi:10.1016/j.jns.2006.03.018.
- Alagarsamy, S.; Rouse, S.T.; Junge, C.; Hubert, G.W.; Gutman, D.; Smith, Y.; Conn, P.J. NMDA-induced phosphorylation and regulation of mGluR5. Pharmacol Biochem Behav 2002, 73, 299-306, doi:10.1016/s0091-3057(02)00826-2.
- Chen, H.-H.; Liao, P.-F.; Chan, M.-H. mGluR5 positive modulators both potentiate activation and restore inhibition in NMDA receptors by PKC dependent pathway. Journal of Biomedical Science 2011, 18, 19, doi:10.1186/1423-0127-18-19.
- Pietraszek, M.; Gravius, A.; Schäfer, D.; Weil, T.; Trifanova, D.; Danysz, W. mGluR5, but not mGluR1, antagonist modifies MK-801-induced locomotor activity and deficit of prepulse inhibition. Neuropharmacology 2005, 49, 73-85, doi:https://doi.org/10.1016/j.neuropharm.2005.01.027.
- Henry, S.A.; Lehmann-Masten, V.; Gasparini, F.; Geyer, M.A.; Markou, A. The mGluR5 antagonist MPEP, but not the mGluR2/3 agonist LY314582, augments PCP effects on prepulse inhibition and locomotor activity. Neuropharmacology 2002, 43, 1199-1209, doi:10.1016/s0028-3908(02)00332-5.
- Chapman, A.G.; Nanan, K.; Williams, M.; Meldrum, B.S. Anticonvulsant activity of two metabotropic glutamate group I antagonists selective for the mGlu5 receptor: 2-methyl-6-(phenylethynyl)-pyridine (MPEP), and (E)-6-methyl-2-styryl-pyridine (SIB 1893). Neuropharmacology 2000, 39, 1567-1574, doi:10.1016/s0028-3908(99)00242-7.
- Yan, Q.J.; Rammal, M.; Tranfaglia, M.; Bauchwitz, R.P. Suppression of two major Fragile X Syndrome mouse model phenotypes by the mGluR5 antagonist MPEP. Neuropharmacology 2005, 49, 1053-1066, doi:10.1016/j.neuropharm.2005.06.004.
- Kano, M.; Watanabe, T. Type-1 metabotropic glutamate receptor signaling in cerebellar Purkinje cells in health and disease. F1000Res 2017, 6, 416, doi:10.12688/f1000research.10485.1.
- Sarantis, K.; Antoniou, K.; Matsokis, N.; Angelatou, F. Exposure to novel environment is characterized by an interaction of D1/NMDA receptors underlined by phosphorylation of the NMDA and AMPA receptor subunits and activation of ERK1/2 signaling, leading to epigenetic changes and gene expression in rat hippocampus. Neurochem Int 2012, 60, 55-67, doi:10.1016/j.neuint.2011.10.018.
- Laurido-Soto, O.; Brier, M.R.; Simon, L.E.; McCullough, A.; Bucelli, R.C.; Day, G.S. Patient characteristics and outcome associations in AMPA receptor encephalitis. J Neurol 2019, 266, 450-460, doi:10.1007/s00415-018-9153-8.
- Höftberger, R.; van Sonderen, A.; Leypoldt, F.; Houghton, D.; Geschwind, M.; Gelfand, J.; Paredes, M.; Sabater, L.; Saiz, A.; Titulaer, M.J.; et al. Encephalitis and AMPA receptor antibodies: Novel findings in a case series of 22 patients. Neurology 2015, 84, 2403-2412, doi:10.1212/wnl.0000000000001682.
- Peng, X.; Hughes, E.G.; Moscato, E.H.; Parsons, T.D.; Dalmau, J.; Balice-Gordon, R.J. Cellular plasticity induced by anti-α-amino-3-hydroxy-5-methyl-4-isoxazolepropionic acid (AMPA) receptor encephalitis antibodies. Ann Neurol 2015, 77, 381-398, doi:10.1002/ana.24293.
- Schidlitzki, A.; Twele, F.; Klee, R.; Waltl, I.; Römermann, K.; Bröer, S.; Meller, S.; Gerhauser, I.; Rankovic, V.; Li, D.; et al. A combination of NMDA and AMPA receptor antagonists retards granule cell dispersion and epileptogenesis in a model of acquired epilepsy. Scientific Reports 2017, 7, 12191, doi:10.1038/s41598-017-12368-6.

Reviewer 2 Report
This review by Sivakumar, Ghasemi and Schachter is giving a reasonable summery of pharmacological, genetic, and immune mediated modulation of NMDA receptors with respect to the pathophysiology and treatment of epilepsy. Therefore, the authors combine data from clinical and pre-clinical data. The manuscript is well structured and written readably. I have however a few minor concerns that need to be addressed
(1) The authors used reference [11] several times. Since this reference is a review article as well, I would prefer the primary literature.
(2) On Page 3 in line 100-104 the authors wrote: “granule cells in non-sprouted hippocampi were not affected by such treatment.” By superficially reading the given references [21-23] I could not find any evidence for this statement. I would kindly ask the authors to check these publications with respect to the statement.
(3) I’m bothered by different nomenclatures for NMDA subunits used in one manuscript.
(4) Page 7 line 300: Please define Na+-channels. (Voltage gated Na+-channels?). What is meant by “glutamate containing neurons”? GABAergic interneurons contain glutamate as well.
(5) Page 7 section 4.8. In my opinion MK-801 is a special NMDAR antagonist blocking in a use- and voltage-dependent manner. I would kindly suggest the authors to find some references and to mention that fact in the text.
(6) There are some typing errors (doubling of space characters, missing superscript e.g.).
Author Response
Reviewer’s Comments
This review by Sivakumar, Ghasemi and Schachter is giving a reasonable summery of pharmacological, genetic, and immune mediated modulation of NMDA receptors with respect to the pathophysiology and treatment of epilepsy. Therefore, the authors combine data from clinical and pre-clinical data. The manuscript is well structured and written readably. I have however a few minor concerns that need to be addressed
Comment 1. The authors used reference [11] several times. Since this reference is a review article as well, I would prefer the primary literature.
Response.
This reference now appears only once in this manuscript. Original references added to below statements as included below.
“Major excitatory neurotransmitter acting on the ionotropic receptors located at the presynaptic terminal and post-synaptic membrane in the central nervous system (CNS) [1].”
“In the resting state, the channel is blocked by Mg2+ and remains equally permeable to Na and Ca2+ ions [2]”
“Neurotoxicity resulting from overexcitation has been widely speculated to play a major role in epilepsy [3]”
“Moreover, a variety of animals models of seizure and epilepsy have demonstrated alteration of NMDAR expression and protein levels, although the results are variable depending on the NMDAR subunits, brain regions, and animal species assessed. This has been comprehensively reviewed in the literature[4], [4], [5], [6]”
Comment 2. On Page 3 in line 100-104 the authors wrote: “granule cells in non-sprouted hippocampi were not affected by such treatment.” By superficially reading the given references [21-23] I could not find any evidence for this statement. I would kindly ask the authors to check these publications with respect to the statement.
Response.
On checking these publications, we could not find a reference to link above mentioned statement which was hence removed. It was more of an inference as normal controls utilized in these papers did consist of normal granule cells from rat models (no direct mention of non-sprouted hippocampi) which we inferred were non-sprouted hippocampi as sprouting of fibers were pathognomonic for epileptic changes.
Comment 3. I’m bothered by different nomenclatures for NMDA subunits used in one manuscript.
Response.
Different nomenclatures listed as follows:
NMDA receptor complex sub-unit names used synonymously
NR1 (GluN1)
NR2A (GluN2A)
NR2B (GluN2B)
NR3 (GluN3)
GRIN2A, GRIN2B, GRIN3, are genes encoding GLuN sub-units respectively
We proofread the manuscript to ensure the above nomenclatures were present in the manuscript.
Comment 4. Page 7 line 300: Please define Na+-channels. (Voltage gated Na+-channels?). What is meant by “glutamate containing neurons”? GABAergic interneurons contain glutamate as well.
Response.
Changed Na+ channels to “voltage-gated Na+ currents”. Rephrased sentence as follows with change to hippocampal neurons to be more specific. We wanted to point to differential effects of glutamate over GABA release in hippocampal neurons and misconstrued the statement.
“Presynaptic release of glutamate was reduced by inhibition of voltage-gated Na+ currents in hippocampal neurons”
Comment 5. Page 7 section 4.8. In my opinion MK-801 is a special NMDAR antagonist blocking in a use- and voltage-dependent manner. I would kindly suggest the authors to find some references and to mention that fact in the text.
Response.
Thank you for this insightful comment. The below fact has been stated in the text.
“MK-801 is a special NMDAR antagonist due to well demonstrated effects in both use-dependent and voltage-dependent manner through blockade of ion permeation [7-10]. “
Comment 6. There are some typing errors (doubling of space characters, missing superscript e.g.).
Response.
Corrections made to Lines 47, 59, 79, 82, 179, 205, 215, 263, 331, 349, 361, 446, 485.
References:
- Paoletti, P. Molecular basis of NMDA receptor functional diversity. European Journal of Neuroscience 2011, 33, 1351-1365.
- Lau, C.G.; Takeuchi, K.; Rodenas-Ruano, A.; Takayasu, Y.; Murphy, J.; Bennett, M.V.; Zukin, R.S. Regulation of NMDA receptor Ca2+ signalling and synaptic plasticity. Biochem Soc Trans 2009, 37, 1369-1374, doi:10.1042/bst0371369.
- Barker-Haliski, M.; White, H.S. Glutamatergic Mechanisms Associated with Seizures and Epilepsy. Cold Spring Harb Perspect Med 2015, 5, a022863, doi:10.1101/cshperspect.a022863.
- Jeon, A.R.; Kim, J.E. PDI Knockdown Inhibits Seizure Activity in Acute Seizure and Chronic Epilepsy Rat Models via S-Nitrosylation-Independent Thiolation on NMDA Receptor. Front Cell Neurosci 2018, 12, 438, doi:10.3389/fncel.2018.00438.
- Liu, S.; Liu, C.; Xiong, L.; Xie, J.; Huang, C.; Pi, R.; Huang, Z.; Li, L. Icaritin Alleviates Glutamate-Induced Neuronal Damage by Inactivating GluN2B-Containing NMDARs Through the ERK/DAPK1 Pathway. Front Neurosci 2021, 15, 525615, doi:10.3389/fnins.2021.525615.
- Zhu, J.M.; Li, K.X.; Cao, S.X.; Chen, X.J.; Shen, C.J.; Zhang, Y.; Geng, H.Y.; Chen, B.Q.; Lian, H.; Zhang, J.M.; et al. Increased NRG1-ErbB4 signaling in human symptomatic epilepsy. Sci Rep 2017, 7, 141, doi:10.1038/s41598-017-00207-7.
- Song, X.; Jensen, M.; Jogini, V.; Stein, R.A.; Lee, C.H.; McHaourab, H.S.; Shaw, D.E.; Gouaux, E. Mechanism of NMDA receptor channel block by MK-801 and memantine. Nature 2018, 556, 515-519, doi:10.1038/s41586-018-0039-9.
- Dravid, S.M.; Erreger, K.; Yuan, H.; Nicholson, K.; Le, P.; Lyuboslavsky, P.; Almonte, A.; Murray, E.; Mosely, C.; Barber, J.; et al. Subunit-specific mechanisms and proton sensitivity of NMDA receptor channel block. J Physiol 2007, 581, 107-128, doi:10.1113/jphysiol.2006.124958.
- Xi, D.; Zhang, W.; Wang, H.-X.; Stradtman, G.G., III; Gao, W.-J. Dizocilpine (MK-801) induces distinct changes of N-methyl-d-aspartic acid receptor subunits in parvalbumin-containing interneurons in young adult rat prefrontal cortex. International Journal of Neuropsychopharmacology 2009, 12, 1395-1408, doi:10.1017/S146114570900042X.
- Sato, K.; Morimoto, K.; Okamoto, M. Anticonvulsant action of a non-competitive antagonist of NMDA receptors (MK-801) in the kindling model of epilepsy. Brain Research 1988, 463, 12-20, doi:https://doi.org/10.1016/0006-8993(88)90521-5.

Reviewer 3 Report
In the review article entitled "Targeting NMDA Receptor Complex in Management of Epilepsy" the Authors wanted to present the newest research in which the pathologic role of NMDARs in epilepsy was examined and prepare an overview on NMDAR antagonists that were tested as anticonvulsive agents in both clinical and preclinical studies. Although in the Manuscript the Authors cite as many as 174 articles, only about 70 of them come from the last 10 years, and only less than 40 from the last 5 years. Therefore, in my opinion Authors should revise the list of references and select the most recent ones, and supplement the Manuscript with the latest research works.
Author Response
Reviewer’s Comments
In the review article entitled "Targeting NMDA Receptor Complex in Management of Epilepsy" the Authors wanted to present the newest research in which the pathologic role of NMDARs in epilepsy was examined and prepare an overview on NMDAR antagonists that were tested as anticonvulsive agents in both clinical and preclinical studies. Although in the Manuscript the Authors cite as many as 174 articles, only about 70 of them come from the last 10 years, and only less than 40 from the last 5 years. Therefore, in my opinion Authors should revise the list of references and select the most recent ones and supplement the Manuscript with the latest research works.
Response.
We thank the reviewers for their thorough review and for sharing insightful comments to further enhance this manuscript. We performed a thorough literature review to identify pertinent articles of interest pertaining to targeting NMDA in epilepsy, and hence this comes up as a limitation as we have iterated within the manuscript that more clinical trials are warranted to study individual agents more extensively. Recent shift in paradigm towards precision medicine approaches has made genetic mutations a topic of interest which we did include. We wanted to present articles more relevant to this special issue and journal of interest which further limited dissemination of work. Overall, we have added about 45 new references and supplemented the manuscript with latest research we felt were apt.

Round 2
Reviewer 1 Report
This manuscript incorporates many of the criticism raised by the reviewers. It is too long and many of the info could be incorporated into tables (ie NMDA regulation or pharmacology). In addition the authors should pay particular attention to blank (and often mistaken statements) such as:
1) Line 57-59 Ionotropic twice same sentence
2) Line 77-78 please provide ref. This is a review!
3) line 12-122
4) expression of D-serine? Not sure what this means.
5) Line 152-153 mGluR5 drugs may have many problems but cannot elicit seizures from the cerebellum
Author Response
Reviewer’s Comments
This manuscript incorporates many of the criticism raised by the reviewers. It is too long and many of the info could be incorporated into tables (i.e., NMDA regulation or pharmacology).
Response. We thank the reviewer for his/her previous comments and feedback on our manuscript which markedly improved the quality of the paper in the current format. As the regulation of NMDA and its pharmacology is a complex issue, we tried to describe it in the main manuscript as the text which could be easier for readers of the article. Provision of figure 1 could also help in this regard. Therefore, we did not add any more tables to keep the manuscript as concise as possible.
In addition, the authors should pay particular attention to blank (and often mistaken statements) such as:
Comment 1. Line 57-59 Ionotropic twice same sentence.
Response. This sentence was corrected on lines 57-59.
Comment 2. Line 77-78 please provide ref. This is a review!
Response. The reference was added.
Comment 3. line 12-122 expression of D-serine? Not sure what this means.
Response. This was corrected to “D-serine levels” on line 120.
Comment 4. Line 152-153 mGluR5 drugs may have many problems but cannot elicit seizures from the cerebellum
Response. We thank the reviewer for this important comment. We agree that mGluR5 antagonists may have good effects on seizure threshold, but the overall use of mGluRs antagonists in general could be limited due to their adverse effects on the cerebellar function (e.g., causing ataxia). This was added on line 151-152. We agree that mGluRs agonism may elicit seizure, but not from cerebellum.

Reviewer 3 Report
The authors responded to my comments. The manuscript may be published in its current form.
Author Response
Reviewer’s Comments
The authors responded to my comments. The manuscript may be published in its current form.
Response.
We thank the reviewer for his/her comment, and we are glad that the revised version of our manuscript was satisfying.
